# Isolation and Molecular Characterization of Indigenous *Penicillium chrysogenum*/*rubens* Strain Portfolio for Penicillin V Production

**DOI:** 10.3390/microorganisms11051132

**Published:** 2023-04-26

**Authors:** Amol M. Sawant, Vishwambar D. Navale, Koteswara Rao Vamkudoth

**Affiliations:** 1Biochemical Sciences Division, CSIR-National Chemical Laboratory, Pune 411008, India; am.sawant@ncl.res.in (A.M.S.);; 2Academy of Scientific and Innovative Research (AcSIR), Ghaziabad 201002, India

**Keywords:** *Penicillium chrysogenum*/*rubens*, internal transcribed spacer region, β-tubulin gene, phenoxymethyl penicillin, high-performance liquid chromatography

## Abstract

Beta (β)-lactam antibiotic is an industrially important molecule produced by *Penicillium chrysogenum*/*rubens*. Penicillin is a building block for 6-aminopenicillanic acid (6-APA), an important active pharmaceutical intermediate (API) used for semi-synthetic antibiotics biosynthesis. In this investigation, we isolated and identified *Penicillium chrysogenum*, P. rubens, P. brocae, P. citrinum, Aspergillus fumigatus, A. sydowii, Talaromyces tratensis, Scopulariopsis brevicaulis, P. oxalicum, and P. dipodomyicola using the internal transcribed spacer (ITS) region and the β-tubulin (*BenA*) gene for precise species identification from Indian origin. Furthermore, the *BenA* gene distinguished between complex species of *P. chrysogenum* and *P. rubens* to a certain extent which partially failed by the ITS region. In addition, these species were distinguished by metabolic markers profiled by liquid chromatography–high resolution mass spectrometry (LC-HRMS). Secalonic acid, Meleagrin, and Roquefortine C were absent in *P. rubens*. The crude extract evaluated for PenV production by antibacterial activities by well diffusion method against *Staphylococcus aureus* NCIM-2079. A high-performance liquid chromatography (HPLC) method was developed for simultaneous detection of 6-APA, phenoxymethyl penicillin (PenV), and phenoxyacetic acid (POA). The pivotal objective was the development of an indigenous strain portfolio for PenV production. Here, a library of 80 strains of *P. chrysogenum*/*rubens* was screened for PenV production. Results showed 28 strains capable of producing PenV in a range from 10 to 120 mg/L when 80 strains were screened for its production. In addition, fermentation parameters, precursor concentration, incubation period, inoculum size, pH, and temperature were monitored for the improved PenV production using promising *P. rubens* strain BIONCL P45. In conclusion, *P. chrysogenum*/*rubens* strains can be explored for the industrial-scale PenV production.

## 1. Introduction

Fungi are the diverse group of eukaryotic organisms ubiquitous in nature [1]. The genus *Aspergillus*, *Fusarium*, and *Penicillium* find applications in various fields including food, agricultural, and pharmaceutical industries [2]. The *Penicillium* genus is a filamentous fungus comprising more than 350 species that produce various industrially important molecules such as penicillin, griseofulvin, mycophenolic acid, andrastin A, cephalosporins, chrysogine viridicatol, kojic acid sorbicillin, meleagrin, roquefortine C, xanthocillin, secalonic acid D, F, lumpidin, compactin, gibberellins, and indoleacetic acid [3,4,5,6,7,8,9,10].

Precise microorganism identification is crucial in natural product research. However, understanding fungi is a challenging task which solely relies on macromorphological and micromorphological approaches which may result in incorrect identifications [11]. Multigene approaches are necessary for accurate identifications at the species and strain levels [12,13]. Although the internal transcribed spacer (ITS) region of rDNA has been recommended as an official molecular marker for most fungal classification [14,15], it has failed to distinguish closely related species and clades [4]. Due to their impediments, β-tubulin (*BenA*) and translational elongation factor 1α (*TEF-1α*) are the recognized molecular markers for species-level identification of *Penicillium* and *Fusarium* species, respectively, that can segregate the closely related species.

Penicillin was a breakthrough drug in medical history produced by *P. chrysogenum*/*rubens* species. Benzylpenicillin (Penicillin G; PenG) and phenoxymethyl penicillin (Penicillin V; PenV). Natural penicillins display excellent activity against both Gram-positive and selected Gram-negative bacteria [16]. Natural penicillins are the substrate for 6-amino penicillanic acid (6-APA) by penicillin acylases, which is important to produce semisynthetic β-lactam antibiotics. Global demand for semisynthetic β-lactam antibiotics is 6000 metric tons per year, accounting for approximately 65% of the total antibiotic market [17,18]. PenV is formed when Phenoxy acetic acid (POA) is added into a fermentation medium along with a highly stable at acidic pH, whereas PenG is less stable. Hence, it can be used in oral demonstrations to treat especially, strep throat, otitis, and cellulitis, and helps in preventing rheumatic fever [19]. Widespread uses of these β-lactam antibiotics lead to antimicrobial resistance (AMR) globally. AMR is a sign of serious threat to public health which forced researchers to ascertain new and amended antibiotics [20]. Pharmaceutical production and trade have been disrupted during the COVID-19 pandemic in many countries worldwide. In consequence, the use of precise antibiotic for infections is necessary for curing infection and reducing AMR. Hence, domestic production of antibiotics/medicines/Active Pharmaceutical Intermediates (APIs) is necessary to replace the bulk of these APIs.

We have explored the preparation of an indigenous *P. rubens* strain portfolio for industrial scale PenG/V production. In this study, we isolated 109 *Penicillium* isolates from various habitats in India. The obtained strains were evaluated by morphological and microscopic examination for genus confirmation. The results were also authenticated with the help of two molecular markers, the ITS and *BenA* genes. The metabolic taxonomic marker was profiled by liquid chromatography–high resolution mass spectrometry (LC-HRMS). A high-performance liquid chromatographic (HPLC) method was developed for the simultaneous detection of PenV, 6-APA, and POA, which are part of the fermentation system. All isolates were assessed for quantitative PenV production in the submerged fermentation process, quantified by HPLC. 

## 2. Materials and Methods

### 2.1. Chemicals and Reagents

All the media, medium ingredients, ethyl acetate, amyl acetate, acetonitrile, methanol, and dichloromethane were purchased from HiMedia (Mumbai, India). PenV, POA, 6-APA, and phosphoric acid (H_3_PO_4_) were purchased from Sigma Aldrich (Bangalore, India), and formic acid was purchased from Tokyo Chemical Industry (TCI) Chemicals (India).

### 2.2. Isolation and Morphological Characterization of Fungi

Various food grains, poultry feed, soil, and environmental samples were collected from different sites of Maharashtra, Gujarat, Andhra Pradesh, and Telangana, India. The obtained samples were processed for isolation of fungi using dilution and blotter method and incubated at 25 ± 2 °C for 5 days [12]. Pure colonies were preserved on the Potato Dextrose Agar (PDA) slants at 4 °C for further use. The isolated colonies were evaluated for species identification using traditional characterization of the colony texture on specific media such as Czapek Yeast Autolysate (CYA), Malt Extract Agar (MEA), Yeast Extract Supplemented (YES), and Creatine Sucrose (CREA). In addition, surface, reverse colony, and microscopic observations including phialides, spore, and conidial arrangement were performed [12]. 

### 2.3. Molecular Identification of Fungi

A precise identification of fungal species was conducted using dual molecular markers, i.e., ITS region [14] and *BenA* protein-coding gene [21]. Isolated pure fungal species were grown in CYA broth for five days, and grown mycelia were harvested and lyophilized using cold liquid nitrogen. About 100 mg of the powdered mycelium was used for genomic DNA (gDNA) isolation using DNeasy Plant Mini Kit (QIAGEN, Delhi, India, Pvt. Ltd.) according to manufacturer’s instructions. The purity and concentration of DNA was determined using Nanodrop ND 1000 UV–Vis spectrophotometer (Thermo Scientific, Waltham, MA, USA).

#### 2.3.1. ITS Region Amplification

A Polymerase chain reaction (PCR) was performed in a Thermocycler (Eppendorf, Hamburg, Germany) with gDNA (1 µL), GoTaq Ready-mix (5 µL) (Promega, Mumbai, India), forward and reverse primers (0.25 µL) and total volume was made 10 µL with nuclease-free water. The PCR condition was programmed for 35 cycles as follows: initial denaturation at 94 °C (3 min), denaturation at 94 °C (30 s), annealing at 40 °C (1 min), elongation at 72 °C (1 min) and final elongation at 72 °C. After successful PCR, the amplified product was resolved onto agarose gel electrophoresis (1%) for gene amplification. 

#### 2.3.2. *BenA* Gene Amplification

For *BenA* gene amplification, a PCR reaction was set up as explained above. The PCR conditions included initial denaturation at 95 °C (5 min), denaturation at 95 °C (1 min), annealing at 55 °C (55 s), elongation at 72 °C (1 min), and final elongation at 72 °C (10 min). PCR was run for 35 cycles. After successful PCR, the amplified product was resolved onto agarose gel electrophoresis (1%) for gene amplification. 

### 2.4. ITS and BenA Gene Sequence Analysis 

The positive amplicon of both markers was purified using DNA purification kit (Macherey-Nagel, GmbH, Duren, Germany) according to the manufacturer’s instructions. The amplified PCR product was sent for sequencing to Eurofins Genomics India Pvt. Ltd. (Bangalore, India). Obtained sequences were searched in the National Centre for Biotechnology Information (NCBI) database for the highest sequence similarity and total scores using a basic local alignment tool (BLAST), and strain names were noted for further reference. The evolutionary relationship was analyzed using the Neighbor-joining tree (NJ) method on the maximum composite likelihood model by MEGA 7 software, version 7.1. The bootstrap tree constructed from 1000 replicates and percentage taxa coverage are shown next to the branch.

### 2.5. Metabolic Profiling Using LC-HRMS 

Secondary metabolites are a very efficient tool in the species-level identification in the fungal taxonomy. In this investigation, metabolic profiling was performed using LC-HRMS to differentiate *P. chrysogenum* and *P. rubens* [22]. With minor modification, isolated *Penicillium* species were grown onto CYA plates for 7 days at 25 °C. At the end of the incubation period, the agar plugs (~6 mm) were taken from the middle of the colony and transferred to the 10 mL screw-cap glass bottle and extracted twice with ethyl acetate, methanol (MeOH), dichloromethane (3:2:1), and formic acid (1%). The organic fraction was concentrated and eluted into MeOH. The final elute was subjected to the further purification step using Solid Phase Extraction (SPE) cartridge Oasis HLB (Waters, Milford, MA, USA). Briefly, the cartridge was equilibrated using 1 mL MeOH:water (50:50) before the purification procedure. Next, the cartridge was washed with 1 mL of MeOH:W=water (5:95), applied to the cartridge and kept stable until the cartridge became completely dried. Finally, the samples were eluted into 1 mL of MeOH and characterized using LC-HRMS [13]. Briefly, a gradient method consisting of acetonitrile (ACN) with 0.1% formic acid (C) and water with 0.1% formic acid (D). The gradient was first at 2% ©, and 98% (D) for 0–30 s. Next, from 30 s to 10 min (C), it changed from 2% to 45% and (D) from 98% to 55%. From 10 to 13 min (C), it changed from 45% to 98% and (D) from 55% to 2%, which was then changed to 2% (C) and 98% (D) in 30 s and held at the same concentration until the next sample injection. The sample (2 µL) was injected and scanned between 100 and 2000 Da for 15 min. Data acquisition and processing were conducted by X Calibur^TM^ software, version 4.0 (Thermo Scientific, Waltham, MA, USA).

### 2.6. Biosynthesis of PenV Using Submerged Fermentation 

*Penicillium chrysogenum*/*P. rubens* are well known for β-lactam antibiotic (PenV/G) production. These isolated species were evaluated for PenV production in the submerged fermentation process. Briefly, 7-day-old *P. chrysogenum*/*P. rubens* spore suspension (1 × 10^8^/mL) was prepared using a Neubauer chamber, inoculated into 10 mL of seed medium containing (g/L); KCl—10, Glucose—20, Yeast nitrogen base—6.6, Citric acid—1.5, K_2_HPO_4_—6, Yeast extract—2, and incubated at 25 °C under 180 rpm in a shaker incubator (Hi-Point, Kaohsiung, Taiwan) for 24 h. Following incubation, the prepared seed inocula were transferred into 40 mL of newly defined penicillin-producing medium containing (g/L); Glucose—1, Lactose—20, Yeast extract—10, Corn steep liquor—5, Beef extract—0.075, Peptone—0.125, (NH_4_)_2_SO_4_—4, KH_2_PO_4_—3, ZnSO_4_·7H_2_O—0.01, MgSO_4_·7H_2_O—2.3, POA—1, and incubated at 25 °C under 180 rpm for 10 days. After incubation, cultures were harvested and extracted for PenV production. 

### 2.7. Optimization of Fermentation Parameters for PenV Production

The effect of various fermentation parameters including POA concentration (0.05 to 0.10) inoculum size (1 × 10^6^, 1 × 10^7^ and 1 × 10^8^, and 1 × 10^9^ spores/mL), temperature (20, 25, 30, 35, and 40 °C), initial pH of the medium (2–11) and incubation period (10 days) on the growth of *P. chrysogenum* BIONCL P45 and PenV production were examined. The culture was grown in a 100 mL working volume in a 500 mL Erlenmeyer flask and incubated as described earlier for improved PenV production. After every 24 h of incubation, 10 mL of the sample was withdrawn and analyzed for PenV production by the HPLC method (described in Section 2.10). All the experiments were carried out in triplicate.

### 2.8. Downstream Processing (DSP) of PenV

At the end of the incubation period, the fermented broth was harvested by filtering the fungal biomass using filter paper, and the obtained cell-free supernatant (CFS) was used for the DSP of PenV [23]. With minor modifications, the CFS was chilled to 4 °C for 30 min, and pH was adjusted to 2.5 with H_2_SO_4_. Subsequently, an equal volume of chilled n-butyl acetate was added and extracted to the acidified broth with horizontal shaking for 5 min. Further, the organic fraction was separated, and an equal volume of cold 10 mM phosphate buffer (pH 7.5) was added. Finally, the aqueous fraction was collected in a clean tube and 1 mL of 0.02% calcium carbonate slurry was added to extract the PenV calcium salt and used for quantitative determination by HPLC. Furthermore, the recovery studies were also conducted by artificially spiking 1% standard PenV in an aqueous sample (Milli Q) and a newly defined penicillin-producing medium for 7 days. At the end of the incubation period, DSP was performed with an equal volume of n-butyl acetate (*n* 3), as discussed above.

### 2.9. Antibiotic Sensitivity Assay for PenV Production

A routine antibiotic sensitivity test was performed to check the production of PenV from the fermented broth of test strains. Antimicrobial sensitivity assay was performed against the pre-grown culture of *Staphylococcus aureus* NCIM-2079 on Muller–Hinton Agar (MHA) plates. A well of 6 mm in diameter was made using a sterile borer, and 50 µL of filtered and extracted broth was added to each well. Positive control was maintained with varying concentrations from 1 µg/mL to 50 µg/mL of standard PenV. Thus, prepared plates were incubated at 37 °C for 18–24 h in an incubator (Genaxy Scientific, New Delhi, India). Following incubation, the zone of inhibition was measured by the Hi antibiotics zone scale (HiMedia, Mumbai, India), and the relative PenV concentration was calculated by comparing it with standard PenV. 

### 2.10. Development of HPLC Method for Detection of PenV, POA, and 6-APA 

In this study, the HPLC method for simultaneous detection of 6-APA, PenV, and POA molecules was developed using C18 analytical column X-Bridge, 4.6 × 250 mm in size with 5 µm particle size, 1525 binary pump, 2489 UV–Visible detector, and 2707 autosampler system (Waters, USA). The mobile phase consisting of MeOH: water, ACN: water, and MeOH: ACN: water with varying concentrations and pH were tested for better separation of these molecules. The chromatography analysis was processed by injecting 20 µL of the test sample with a flow rate of 1 mL/min and detected at 210 nm. The separation and quantification of the test sample were compared with the standard graph prepared with standard molecules, 6-APA, PenV, and POA. The limit of detection (LOD), the limit of quantification (LOQ), and the precession of the method were measured by preparing calibration curves of commercial standards. 

## 3. Results

### 3.1. Isolation and Morphological Identification of Fungal Species

In this study, we aim to landscape an indigenous *P. chrysogenum*/*rubens* strain portfolio for PenV production. We have collected various samples for the isolation of *Penicillium* species and authenticated them on *Penicillium* specific media. *Penicillium* species have been identified using traditional approaches such as aerial, reverse colony color and morphology, conidium, colonial ornamentation, pigment formation, and growth rate on CYA medium (Appendix A). We have isolated various species belonging to the genera *Penicillium*, *Aspergillus*, *Scopulariopsis*, and *Talaromyces* based on colonies similar to *Penicillium* morphology as selection criteria. As a result, we obtained 109 isolates belonging to the *P. chrysogenum*, *P. rubens*, *P. citrinum*, *P. brocae*, *P. oxalicum*, *P. dipodomyicola*, *A. sydowii*, *A. fumigatus* and *T. tratensis* species. *Penicillium* species were determined to be the dominant genus among them (Table 1).

The phenotypic differentiation of closely related *Penicillium* species, mostly *P. chrysogenum* and *P. rubens*, is firmly linked, which is difficult to distinguish (Figure 1). The colony colors of most isolates were creamy white, bluish green to light green aerially, and yellow, pale yellow to brownish yellow in reverse. Despite this, interestingly, the differential growth pattern with floccose white growth with dark brown reverse coloration was sown by *P. brocae* BIONCL P81 strain which is similar to *A. sydowii* BIONCL P95 isolated from soil. The different geographical locations and habitats may cause variations in the growth pattern.

### 3.2. Molecular Identification by ITS Regions

The ITS and *BenA* gene markers have been used to identify fungal species. Following successful gene amplification, all the strains were chosen for the BLASTn analysis at NCBI. The nucleotide sequences displayed with high query coverage and sequence similarity (to 98%) were assigned the same species name. A total of 109 isolates were analyzed with ITS marker, out of which 40 isolates showed the closest similarity with *P. chrysogenum*, and 34 isolates were similar to *P. rubens*. The rest of the isolates belong to *P. citrinum*, *P. brocae*, *P. oxalicum*, *P. dipodomyicola*, *A. sydowii*, *A. fumigatus* and *T. tratensis* (Figure 2).

### 3.3. Molecular Identification by BenA Gene

All the isolates were analyzed with the *BenA* gene for precise identification of closely related species complexes; 39 isolates showed the closest similarity with *P. chrysogenum* and 41 with *P. rubens*. *Penicillium oxalicum* and *P. dipodomyicola* strains from ITS identification showed similarity with *P. rubens* when analyzed with the *BenA* gene. Other isolates from *Aspergillus*, *Scopulariopsis*, and *Tratensis* genera fall under the same genus with *BenA* gene identification. ITS failed to differentiate between the closely related strains of the *Penicillium* genus to some extent. Furthermore, a phylogenetic tree was constructed for all isolates analyzed and revealed that they clustered in groups with a close resemblance. Concerning *P. chrysogenum* and *P. rubens* differentiation, the *BenA* marker gene showed a distinct grouping formation compared to the ITS marker (Figure 3). The *BenA* gene showed a separate grouping from the ITS region within the *Penicillium* genus, mainly *P. chrysogenum* and *P. rubens*.

### 3.4. Metabolic Profiling by LC–HRMS

Metabolic profiling is a recognizable proof in fungal taxonomy for the precise segregation of species. In this investigation, one representative strain of *P. chrysogenum* and *P. rubens* was profiled by LC-HRMS. Roquefortine C, chrysogine, sorbicillin, meleagrin, andrastin A, xanthocillin X, secalonic acid D, lumpedin, and penicillin are produced by the *P. chrysogenum* (Figure 4). Moreover, *P. rubens* can produce all metabolites except secalonic acid, meleagrin, and roquefortine C. The results suggest that *P. chrysogenum* strain can be distinguished from *P. rubens* using metabolite production. Nevertheless, both strains showed diversity in PenV production and other metabolites.

### 3.5. PenV Production

Isolated and authenticated *Penicillium* strains were screened for PenV production in a newly defined penicillin-production medium. After screening of 80 *P. chrysogenum*/*rubens* strains for PenV production, 28 were determined to be capable of producing 10–100 mg/L of PenV in submerged fermentation, quantified by HPLC (Table 1). Among all screened strains, *Penicillium rubens* strain BIONCL P45 was determined to be the highest PenV producer. This strain was chosen for further studies for enhanced antibiotics production by optimizing various fermentation and other important parameters.

### 3.6. Optimization Fermentation Parameters

Various fermentation parameters including POA concentration, incubation period, inoculum size, pH, and temperature were monitored for the improved PenV production using *P. rubens* BIONCL P45 strain. Here, POA was used as precursor for biosynthesis of PenV. Moreover, POA has also been used in certain herbicide and antifungal drugs. Hence, the effect of POA concentration on vegetative growth and PenV production by *P. rubens* was studied. Interestingly, no significant impact was observed on vegetative growth (Figure 5). However, biosynthesis of PenV was severely affected. A maximum PenV production was observed when inoculated with 0.01% POA, and biosynthesis of PenV was reduced with increasing and decreasing concentrations (Figure 6A). In addition, at higher concentration, traces of POA remain unutilized and affect DSP and PenV quantification. Results suggest that biosynthesis of PenV began at 72 h of incubation, and on the 5th day, there was a sharp rise in its production. On the 8th day of an extended incubation period, 101 mg/L of product was produced. However, further incubation did not result in an increase in PenV titer (Figure 6B).

Furthermore, the optimal production of PenV was achieved with 1 × 10^8^ spore inoculum. With low number of spores in inoculum, mycelium formed a ball-like pellet structure in the production medium. As inoculum size increases, the growth of mycelia changes from pellets to pulpy with decrease in PenV production, and this might occur because of limitation in nutrients and oxygen supply (Figure 6C). Incubation temperature evaluated from 20 to 40 °C, and *P. rubens* exhibited high PenV production (101 mg/L) at 25 °C, and bellow and above 25 °C it exhibited a sharp reduction in its biosynthesis (Figure 6D). In addition, since initial pH of the medium affects the PenV production, the optimum pH was determined to be 5. Below and above, a decrease in PenV, and at extreme pH values (3 and 9), no cell growth was observed (Figure 6E).

### 3.7. DSP and Recovery of PenV

DSP was performed to harvest the PenV from the whole fermented broth in pure form. To evaluate the percent of recovery, the spiking study of standard 1% PenV was performed in PenV production medium. As mentioned in the methodology, the extraction was performed three times with butyl acetate, and finally, the eluted PenV in aqueous phase was quantified by HPLC. In the first extraction cycle, about 61% recovery rate was observed. Second cycle of extraction was performed with organic phase remaining in first cycle, and the 8% PenV was extracted. Finally, in the third cycle of extraction, about 3% of recovery was observed. In all three cycles, overall, a 72% recovery rate was observed, and 28% of PenV was lost during the extraction, which might be due to the lower pH and the organic solvent used in the process. Finally, PenV was harvested from the fermented broth and quantified by HPLC.

### 3.8. Antibiotic Plate Assay

Antibiotic assay is routinely used to confirm the antimicrobial activity of drugs/molecules. The fermented broth was extracted for PenV production, its activity evaluated against *S. aureus* NCIM-2079. The antimicrobial activity and HPLC results were positively correlated and used to screen the strains for PenV production. The tested strain showed a 10 to 38 mm zone of inhibition (Appendix A). The data of test samples produce 10 to 100 mg/L PenV when compared with standard PenV.

### 3.9. Development of an HPLC Method for the Detection of PenV, POA, and 6-APA

In this investigation, precise HLPC method was developed for simultaneous, quantitative detection of PenV, POA, and 6-APA. Results reported that ACN:water (50:50), pH 3.0 was the most suitable for the distinct separation. Retention times for 6-APA, POA, and PenV were recorded at 2.49, 3.72, and 4.39 min, respectively, with good resolution (Figure 7). The calibration curve was plotted by considering the peak area and the relative concentration (Appendix A), and then the regression equation was computed for each of the molecules; the values are listed in Table 2. Correlation coefficients (R^2^), 0.9988 and 0.9999, were observed for PenV and 6-APA, respectively. However, (R^2^) 0.9373 was determined for POA.

Furthermore, to check the limit of detection (LOD) and limit of quantitation (LOQ), lower concentrations from 1 to 10 µg/mL were tested, and it was determined that the developed method can detect the molecules from 5 to 10 µg/mL and can quantify them from 10 µg/mL. The developed method distinctly separates the PenV, 6-APA, and POA from the mixture quickly and precisely. All the samples were processed in a triplicate to check the reproductivity of the method.

## 4. Discussion

*Penicillium rubens* (formerly named *P. notatum*) was the first penicillin-producing microorganism isolated by Sir Alexander Flaming, and the bioactive molecule penicillin was characterized by Howard Walter Florey and Ernst Boris Chain [24]. Subsequently, findings suggest various species of *Penicillium*, *Aspergillus*, and *Fusarium* which have been characterized for their ability to produce penicillins and their industrially important drugs/molecules [25]. In this investigation, we isolated various strains of *P. chrysogenum*/*rubens* from diverse habitats, such as India [26,27,28]. Isolated species were examined on *Penicillium*-specific media for morphological characterization and distinguishing *Penicillium* species [27,29,30]. Traditionally, colony color, texture, size, spore shape, and exudate formation are the key features used for morphological differentiation of *Penicillium* species and helpful in species differentiation to date [12,13]. Unfortunately, in most cases, morphological and microscopic observations help identifying genus and certain species, but not closely related species, which can be overcome by using molecular markers. Since differentiation of all penicillium species is difficult by morphological observations, multilocous sequence analysis (MLSA) such as ITS, *BenA*, CaM, NL, and LR are really helpful to identify complex species [13]. We have isolated and prepared an indigenous strain portfolio for the *P. chrysogenum*/*rubens* for PenV production. The ITS is the universally accepted official DNA barcode for the fungal taxonomy [15]. Unfortunately, all closely related species of verticillate *Penicillium* and ascomycetes have not differentiated in the ITS barcode. Due to their taxonomic difficulties, protein-coding genes are often required for species-level differentiation, which has been validated by *BenA* marker [31,32,33]. Recent classification suggests two clades representing *P. chrysogenum* and *P. rubens*, and concludes *P. rubens* is the original flaming penicillin-producing strain [4]. In our study, the ITS sequence analysis of 109 strains revealed that about 40 isolates were closely related to *P. chrysogenum*, and 34 strains mapped to *P. rubens*. Furthermore, these strains were analyzed with the *BenA* marker to verify the species identification; unexpectedly, *P. chrysogenum*, *P. oxalicum*, and *P. dipodomyicola* strains showed similarity with *P. rubens* mapped in common clade with *P. chrysogenum* and *P. rubens* [34]. Hence, we have performed several assays and metabolic profiling of reported molecules for distinguishing closely related species [35]. *Penicillium* strains are known for producing important molecules; of note, *P. chrysogenum* strains produce sorbicillin, meleagrin, xanthocillin X, roquefortine C, andrastin A, secalonic acid, penicillin, and chrysogine [36]. On the other hand, no such reports are available for the biosynthesis of secalonic acid, meleagrin, and roquefortine C by *P. rubens*. *P. rubens* clade is best known for penicillin, roquefortine C, chrysogine, meleagrin, sorbicillin, and xanthocillin X [37]. Moreover, *P. chrysogenum* clade produces secalonic acids D, F, and lumpidin-like compounds along with metabolites produced by *P. rubens* strains [4]. *Penicillium* strains are known for producing essential molecules, including enzymes, organic acids, terpenoids, and polyketides used in various industries (Table 3) [38,39,40,41,42,43,44,45,46,47,48,49,50,51,52,53,54,55,56,57,58,59,60,61,62,63,64,65,66,67,68,69,70].

The pivotal aim of the study is the development of candidate *P. chrysogenum*/*rubens* strains for PenV production. The prepared library of 80 strains of both or either *P. chrysogenum*/*rubens* were evaluated for production of PenV. The precise method is important for quantification of the produced molecules in fermentation. Hence, HPLC method is a critical process in identifying different pharmaceutical products [71]. The separation of the structurally related compounds is dependent upon the selection of mobile phase composition, pH, column type, etc. We have developed a precise, sensitive, and reliable liquid chromatographic method for simultaneous quantitative detection of PenV, 6-APA, and POA. Among the studied mobile phases, ACN:water (50:50) and pH 3 with H_3_PO_4_ showed good separation of studied molecules. The combination of ACN and water (60:40) with pH 4 by H_3_PO_4_ was successfully used for the separation of the β-lactam antibiotic and 6-APA [72]. Our developed method showed the distinct peaks of individual molecules in mixture, as well as reproducibility.

In order to enhance the production of PenV, optimization of nutritional parameters is very important [73]. In our study, the POA concentration from 0.1 to 1% was tested for effect on growth, and 0.005 to 0.1% concentration was tested for PenV production. The precursor concentration between 0.1 and 1% did not negatively affect growth. Furthermore, 0.01% POA resulted in higher PenV biosynthesis and further increases in POA concentration there is reduction in PenV titer. Industrial PenV producing strains utilize more than 2.5% of POA [74]. However, all the strains are unable to utilize the higher precursor concentration and may affect growth and PenV production. Conventional penicillin production in a defined medium containing corn steep liquor, glucose, lactose, minerals, oil, and specific precursors enhances the production of the *P. chrysogenum* Q176 strain [75,76,77]. Inoculum size, temperature, incubation period, and initial pH majorly affects the penicillin titer [78]. The optimum inoculum size for higher penicillin production and mycelial growth is 10^7^ spores/dm^3^, but this is not always the same for all the strains [79]. Kumar and co-workers [80] studied the optimum penicillin production at 26 °C; the optimum production temperature varies from 23 to 28 °C, but good mycelium growth was discovered at 30 °C [81,82]. Our results are positively correlated with those reported by Kumar et al. [80] at 25 °C, and the active time of PenV production is 6–8 days [83]. PenV production is very sensitive to the initial pH of the inoculum media, and it was observed that a pH less than 4 or more than 7 was not suitable for growth and product formation [79]. Finally, the studied *P. rubens* BIONCL P45 strain produced the highest PenV at pH 5 on the 8th day with 1 × 10^8^ spores and 0.01% POA concentration. Furthermore, more studies are needed, including classical and genetic engineering approaches, for the development of industrial PenV-producing *P. rubens* strains to ensure bulk API production.

## 5. Conclusions

In the present study, 80 *P. chrysogenum*/*rubens* strains were collected from diverse habitats in India. Dual molecular markers, ITS, and *BenA* barcodes were used for species differentiation. An array of new techniques and molecular and biochemical methods must be improved in the fungal taxonomy of industrial importance. An inherent variability in species of *Penicillium* constitutes a serious problem for taxonomists due to new physiological approaches. Furthermore, 35% of isolated strains were capable of producing PenV at very low levels. Further, repetitive DSP could able to achieve 72% of PenV. The chromatographic method precisely detected the product, precursors, and by-products. A promising strain of *P. rubens* BIONCL P45 was explored for PenV production, and various fermentation parameters were optimized for enhanced PenV production at the lab scale. It can be explored for the industrial-scale production of PenV to substitute imports. More detailed studies are needed to understand the biosynthetic pathway genes, compartmentalization, nutrient uptakes, and transporter proteins for commercial PenV production. 

## Figures and Tables

**Figure 1 microorganisms-11-01132-f001:**
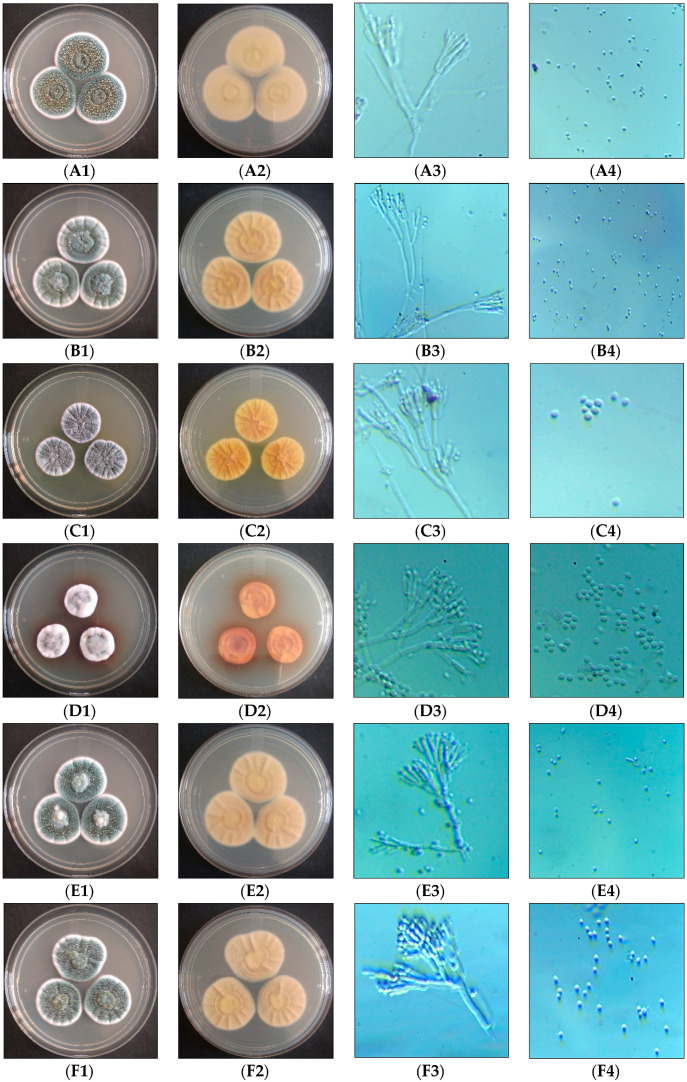
Morphological observation of isolated species. **A**—*P. rubens*, **B**—*P. chrysogenum*, **C**—*P. citrinum*, **D**—*P. brocae*, **E**—*P. oxalicum*, **F**—*P. dipodomyicola*, **G**—*T. tratensis*, **H**—*A. sydowii*, **I**—*A. fumigatus*, **J**—*S. brevicaulis*; **1**—aerial, **2**—revers, **3** and **4**—microscopic observation (Magnification 400×).

**Figure 2 microorganisms-11-01132-f002:**
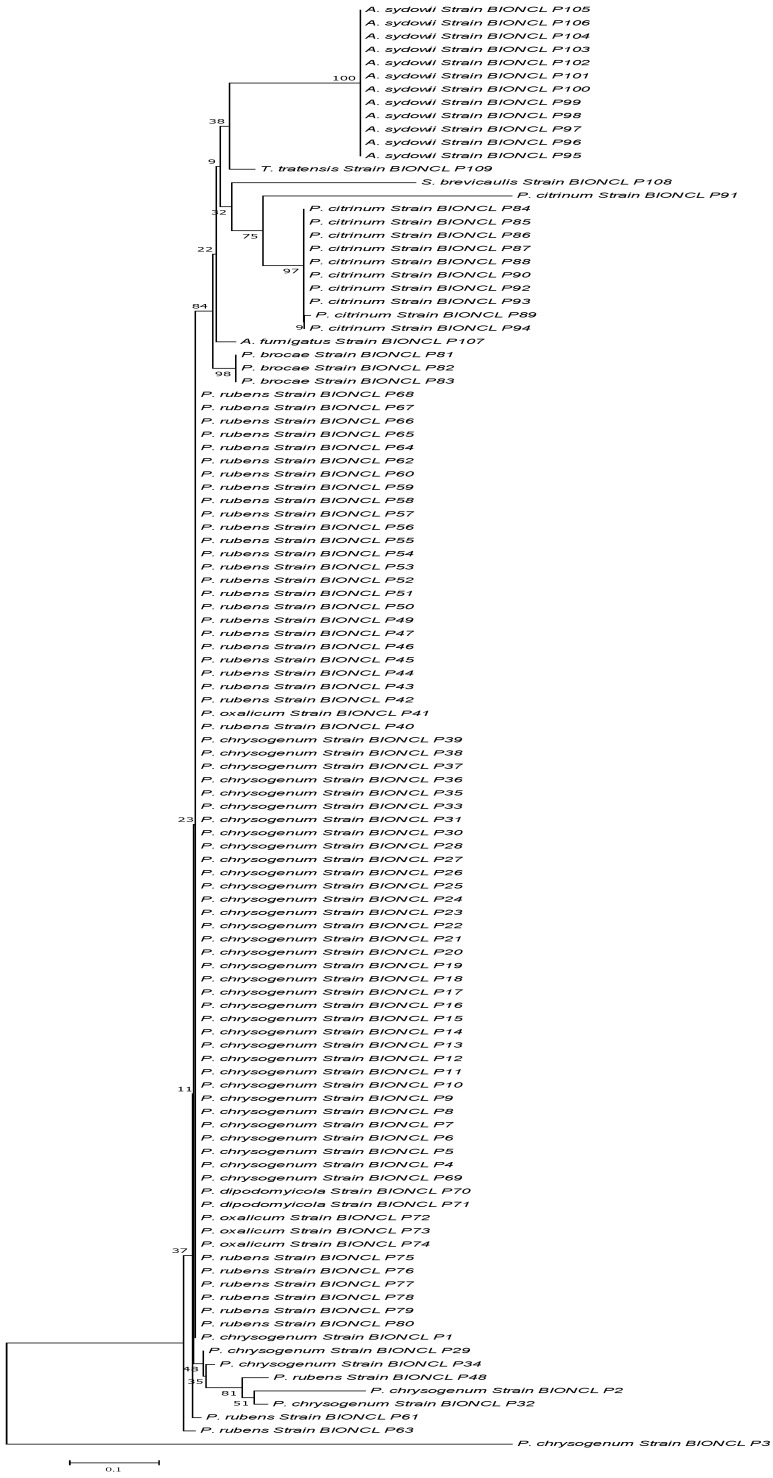
Phylogenetic analysis of the isolates based on the ITS sequencing.

**Figure 3 microorganisms-11-01132-f003:**
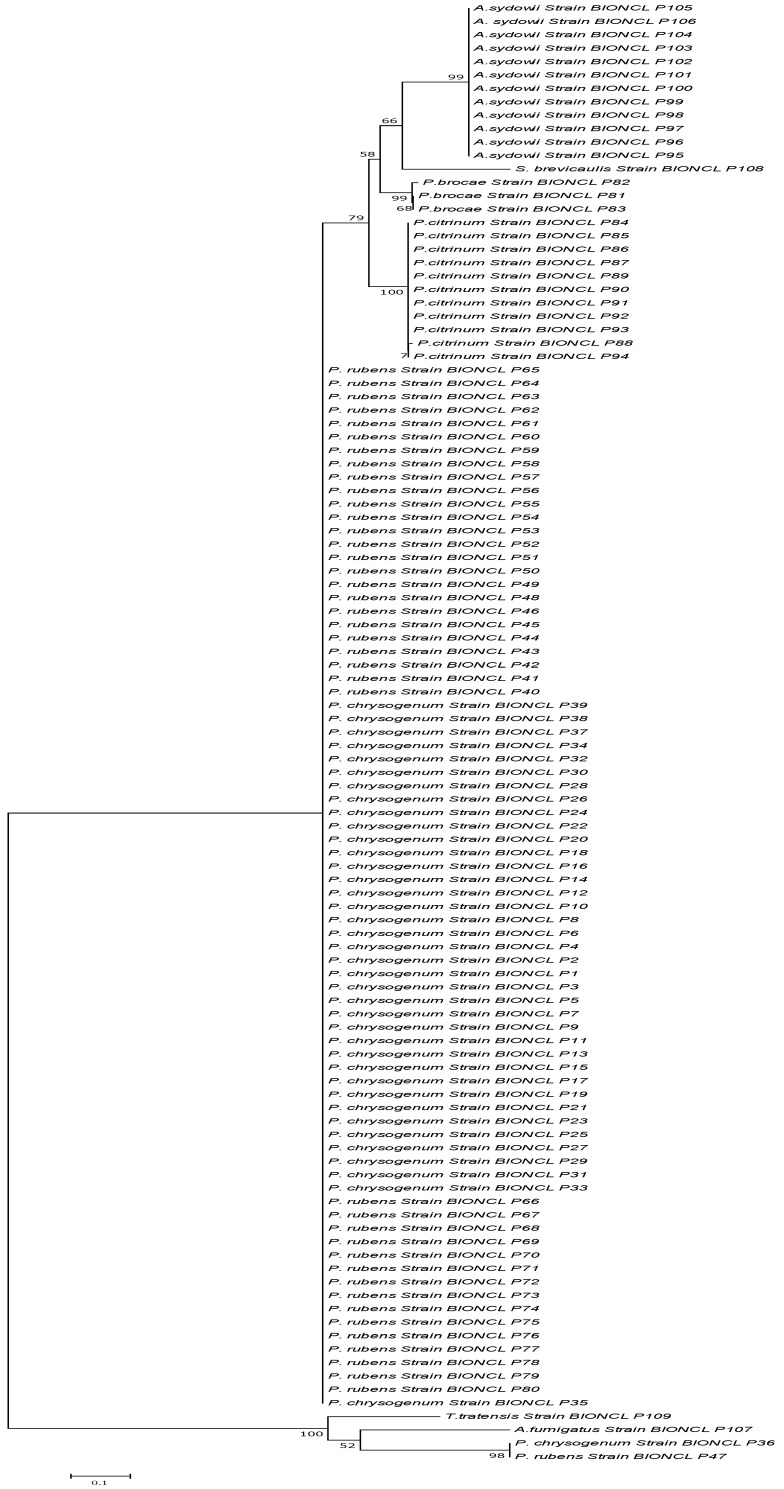
Phylogenetic analysis of the isolates based on the *BenA* marker sequencing.

**Figure 4 microorganisms-11-01132-f004:**
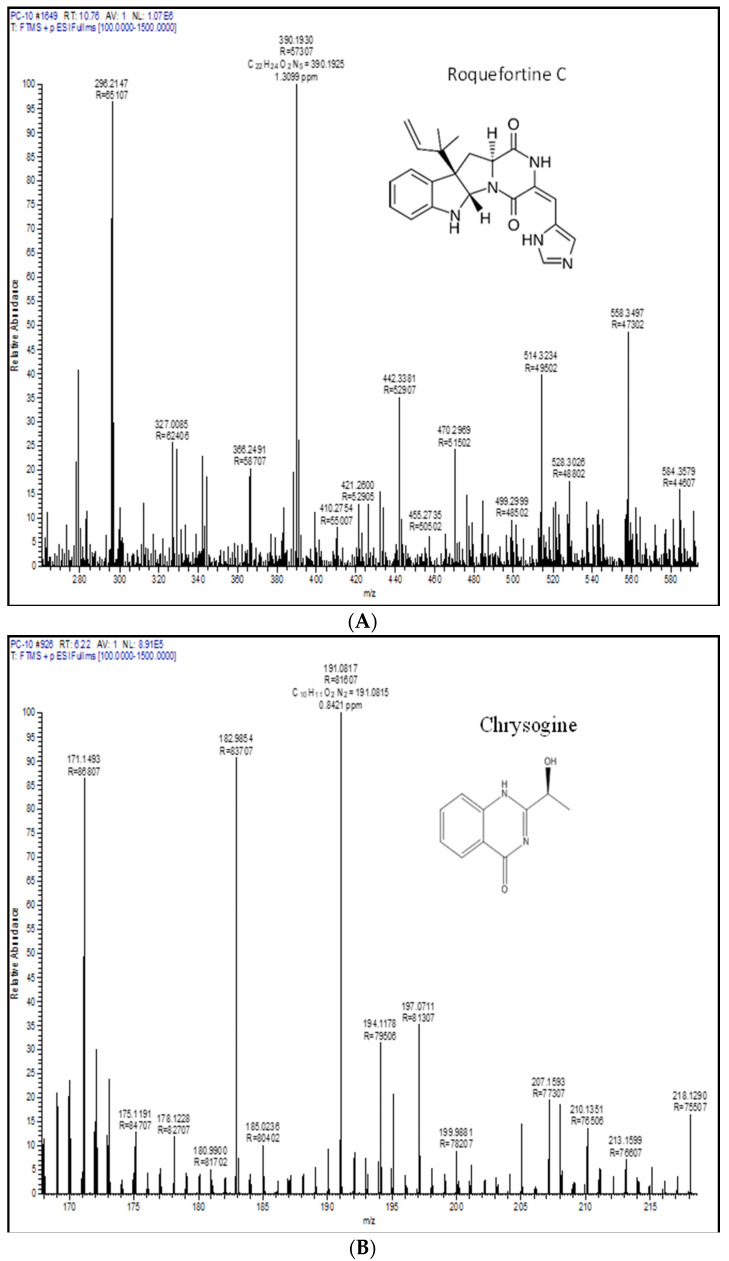
Extrolite analysis of *P. chrysogenum* by LC-HRMS (Q-exactive-orbitrap Mass Spectrometer) with a positive ionization mode. **A**—roquefortine C, **B**—chrysogine, **C**—sorbicillin, **D**—meleagrin, **E**—andrastin A, **F**—xanthocillin X, **G**—secalonic acid, **H**—lumpidin, **I**—penicillin.

**Figure 5 microorganisms-11-01132-f005:**
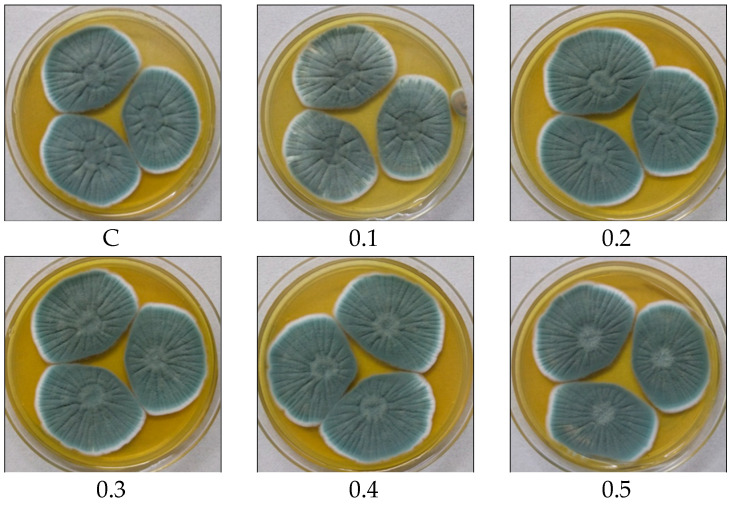
Effect of POA concentration on growth and biomass of *P. rubens* BIONCL P45. C—Control, 0.1 to 1.0 concentration of POA (%).

**Figure 6 microorganisms-11-01132-f006:**
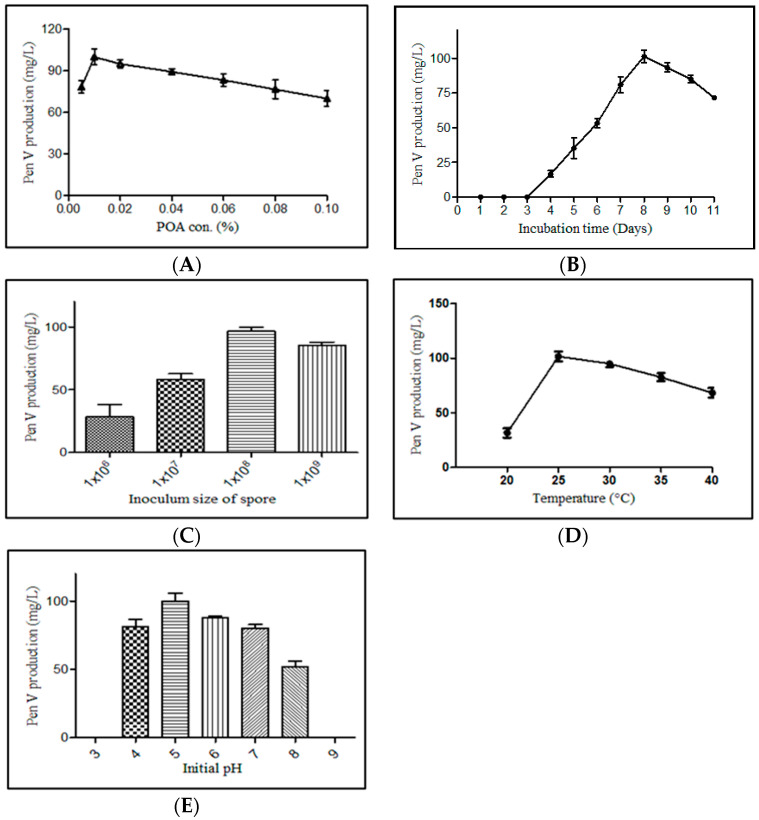
Optimization of fermentation parameters for PenV production by *P. rubens* BIONCL P45. (**A**)—POA concentration optimization, (**B**)—Effect of incubation time on PenV production, (**C**)—Effect of inoculum size on PenV production, (**D**)—Effect of temperature on PenV production, (**E**)—Effect of initial pH on PenV production.

**Figure 7 microorganisms-11-01132-f007:**
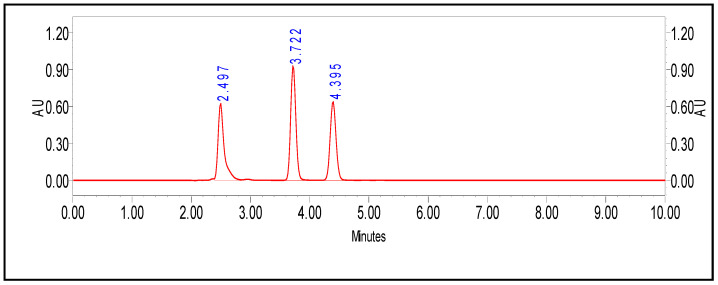
Quantitative analysis of PenV and simultaneous detection of 6-APA, POA, and PenV by HPLC method.

**Table 1 microorganisms-11-01132-t001:** Molecular characterization and screening of *Penicillium chrysogenum*/*rubens* strains and PenV production.

Isolate Name	Molecular Identification		PenV (mg/L)
	ITS	β-tubulin	
BIONCL P1	*P. chrysogenum*	*P. chrysogenum*	48
BIONCL P2	*P. chrysogenum*	*P. chrysogenum*	ND
BIONCL P3	*P. chrysogenum*	*P. chrysogenum*	ND
BIONCL P4	*P. chrysogenum*	*P. chrysogenum*	42
BIONCL P5	*P. chrysogenum*	*P. chrysogenum*	ND
BIONCL P6	*P. chrysogenum*	*P. chrysogenum*	18
BIONCL P7	*P. chrysogenum*	*P. chrysogenum*	16
BIONCL P8	*P. chrysogenum*	*P. chrysogenum*	24
BIONCL P9	*P. chrysogenum*	*P. chrysogenum*	ND
BIONCL P10	*P. chrysogenum*	*P. chrysogenum*	ND
BIONCL P11	*P. chrysogenum*	*P. chrysogenum*	ND
BIONCL P12	*P. chrysogenum*	*P. chrysogenum*	ND
BIONCL P13	*P. chrysogenum*	*P. chrysogenum*	ND
BIONCL P14	*P. chrysogenum*	*P. chrysogenum*	18
BIONCL P15	*P. chrysogenum*	*P. chrysogenum*	ND
BIONCL P16	*P. chrysogenum*	*P. chrysogenum*	ND
BIONCL P17	*P. chrysogenum*	*P. chrysogenum*	3
BIONCL P18	*P. chrysogenum*	*P. chrysogenum*	63
BIONCL P19	*P. chrysogenum*	*P. chrysogenum*	ND
BIONCL P20	*P. chrysogenum*	*P. chrysogenum*	ND
BIONCL P21	*P. chrysogenum*	*P. chrysogenum*	ND
BIONCL P22	*P. chrysogenum*	*P. chrysogenum*	ND
BIONCL P23	*P. chrysogenum*	*P. chrysogenum*	21
BIONCL P24	*P. chrysogenum*	*P. chrysogenum*	ND
BIONCL P25	*P. chrysogenum*	*P. chrysogenum*	ND
BIONCL P26	*P. chrysogenum*	*P. chrysogenum*	ND
BIONCL P27	*P. chrysogenum*	*P. chrysogenum*	ND
BIONCL P28	*P. chrysogenum*	*P. chrysogenum*	ND
BIONCL P29	*P. chrysogenum*	*P. chrysogenum*	ND
BIONCL P30	*P. chrysogenum*	*P. chrysogenum*	32
BIONCL P31	*P. chrysogenum*	*P. chrysogenum*	22
BIONCL P32	*P. chrysogenum*	*P. chrysogenum*	ND
BIONCL P33	*P. chrysogenum*	*P. chrysogenum*	12
BIONCL P34	*P. chrysogenum*	*P. chrysogenum*	ND
BIONCL P35	*P. chrysogenum*	*P. chrysogenum*	24
BIONCL P36	*P. chrysogenum*	*P. chrysogenum*	ND
BIONCL P37	*P. chrysogenum*	*P. chrysogenum*	14
BIONCL P38	*P. chrysogenum*	*P. chrysogenum*	16
BIONCL P39	*P. chrysogenum*	*P. chrysogenum*	ND
BIONCL P40	*P. rubens*	*P. rubens*	43
BIONCL P41	*P. oxalicum*	*P. rubens*	21
BIONCL P42	*P. rubens*	*P. rubens*	24
BIONCL P43	*P. rubens*	*P. rubens*	14
BIONCL P44	*P. rubens*	*P. rubens*	ND
BIONCL P45	*P. rubens*	*P. rubens*	100
BIONCL P46	*P. rubens*	*P. rubens*	ND
BIONCL P47	*P. rubens*	*P. rubens*	ND
BIONCL P48	*P. rubens*	*P. rubens*	ND
BIONCL P49	*P. rubens*	*P. rubens*	11
BIONCL P50	*P. rubens*	*P. rubens*	ND
BIONCL P51	*P. rubens*	*P. rubens*	ND
BIONCL P52	*P. rubens*	*P. rubens*	16
BIONCL P53	*P. rubens*	*P. rubens*	ND
BIONCL P54	*P. rubens*	*P. rubens*	ND
BIONCL P55	*P. rubens*	*P. rubens*	ND
BIONCL P56	*P. rubens*	*P. rubens*	ND
BIONCL P57	*P. rubens*	*P. rubens*	ND
BIONCL P58	*P. rubens*	*P. rubens*	ND
BIONCL P59	*P. rubens*	*P. rubens*	ND
BIONCL P60	*P. rubens*	*P. rubens*	6
BIONCL P61	*P. rubens*	*P. rubens*	18
BIONCL P62	*P. rubens*	*P. rubens*	8
BIONCL P63	*P. rubens*	*P. rubens*	ND
BIONCL P64	*P. rubens*	*P. rubens*	ND
BIONCL P65	*P. rubens*	*P. rubens*	18
BIONCL P66	*P. rubens*	*P. rubens*	ND
BIONCL P67	*P. rubens*	*P. rubens*	ND
BIONCL P68	*P. rubens*	*P. rubens*	17
BIONCL P69	*P. chrysogenum*	*P. rubens*	ND
BIONCL P70	*P. dipodomyicola*	*P. chrysogenum*	ND
BIONCL P71	*P. dipodomyicola*	*P. rubens*	ND
BIONCL P72	*P. oxalicum*	*P. rubens*	ND
BIONCL P73	*P. oxalicum*	*P. rubens*	ND
BIONCL P74	*P. oxalicum*	*P. rubens*	ND
BIONCL P75	*P. rubens*	*P. rubens*	ND
BIONCL P76	*P. rubens*	*P. rubens*	ND
BIONCL P77	*P. rubens*	*P. rubens*	ND
BIONCL P78	*P. rubens*	*P. rubens*	ND
BIONCL P79	*P. rubens*	*P. rubens*	10
BIONCL P80	*P. rubens*	*P. rubens*	ND
BIONCL P81	*P. brocae*	*P. brocae*	NA
BIONCL P82	*P. brocae*	*P. brocae*	NA
BIONCL P83	*P. brocae*	*P. brocae*	NA
BIONCL P84	*P. citrinum*	*P. citrinum*	NA
BIONCL P85	*P. citrinum*	*P. citrinum*	NA
BIONCL P86	*P. citrinum*	*P. citrinum*	NA
BIONCL P87	*P. citrinum*	*P. citrinum*	NA
BIONCL P88	*P. citrinum*	*P. citrinum*	NA
BIONCL P89	*P. citrinum*	*P. citrinum*	NA
BIONCL P90	*P. citrinum*	*P. citrinum*	NA
BIONCL P91	*P. citrinum*	*P. citrinum*	NA
BIONCL P92	*P. citrinum*	*P. citrinum*	NA
BIONCL P93	*P. citrinum*	*P. citrinum*	NA
BIONCL P94	*P. citrinum*	*P. citrinum*	NA
BIONCL P95	*A. Sydowii*	*A. sydowii*	NA
BIONCL P96	*A. Sydowii*	*A. sydowii*	NA
BIONCL P97	*A. Sydowii*	*A. sydowii*	NA
BIONCL P98	*A. Sydowii*	*A. sydowii*	NA
BIONCL P99	*A. Sydowii*	*A. sydowii*	NA
BIONCL P100	*A. Sydowii*	*A. sydowii*	NA
BIONCL P101	*A. Sydowii*	*A. sydowii*	NA
BIONCL P102	*A. Sydowii*	*A. sydowii*	NA
BIONCL P103	*A. Sydowii*	*A. sydowii*	NA
BIONCL P104	*A. Sydowii*	*A. sydowii*	NA
BIONCL P105	*A. Sydowii*	*A. sydowii*	NA
BIONCL P106	*A. Sydowii*	*A. sydowii*	NA
BIONCL P107	*A. fumigatus*	*A. fumigatus*	NA
BIONCL P108	*S. brevicaulis*	*S. brevicaulis*	NA
BIONCL P109	*T. tratensis*	*T. tratensis*	NA

ND—not detected, NA—not applied.

**Table 2 microorganisms-11-01132-t002:** Validation parameters of the developed HPLC method for detection of Pen V, 6-APA, and POA.

Parameter	Pen V	6-APA	POA
Range	100–500 µg/mL	100–500 µg/mL	100–500 µg/mL
Slope	39,169	22,730	27,287
Intercept	925,634	190,701	400,000
R^2^	0.9988	0.9999	0.9373
LOD	5 µg/mL	10 µg/mL	7 µg/mL
LOQ	10 µg/mL	10 µg/mL	10 µg/mL

**Table 3 microorganisms-11-01132-t003:** Metabolites produced by different fungal species and their applications.

Species	Enzyme/Metabolite	Application	References
*P. chrysogenum*	Roquefortine C; Chrysogine; Sorbicillin; Meleagrin; Andrastin A; Xanthocillin X; Secalonic acid D; Lumpidin; Penicillin	Antimicrobial Cheese making	Present study
*P. rubens*	Chrysogine; Sorbicillin; Andrastin A; Xanthocillin X; Lumpidin; Penicillin	Antimicrobial	Present study
*P. brocae*	Fumigatin chlorohydrin; iso-fumigatin chlorohydrin; Penicibrocazine A–E; Brocazines A−G; Spirobrocazine C; Ergosterol; Brocaenol A–C	Antimicrobial, Cytotoxic, Symbiotically associated with insect and provide nutrient	[38,39,40,41,42,43]
*P. citrinum*	Epiremisporine B, D, E; Penicitrinone A; β-glucosidases; Epiremisporine F-H; polyketides and steroids; Cyclopeniol; Gibberellins; Fungal conidia	Anti-inflammatory and cytotoxic; Biomass saccharification; Antifungal; Antiviral; Mosquito controller; Plant growth promoter	[44,45,46,47,48,49,50]
*A. Fumigatus*	β-xylosidase; Cellulase; Amylase; Pectinase; Phosphatase and phytase	Bioethanol production; Textile, detergent, food and feed industries.	[51,52,53,54,55]
*A. sydowii*	Keratinase; Sydowione A–B; Acremolin D; Acremolin C; Lignin and Manganese peroxidase	Degradation of keratinous materials; Antioxidant; Antiproliferative; Antibacterial	[56,57,58,59]
*T. tratensis*	Dry powder formulation; Cell free supernatant	Antagonistic activity against plant diseases and growth promotor	[60,61]
*S. brevicaulis*	Scopularides A–B; Keratinase	Bioremediation of PAH; Degradation of keratinous material; Anticancer and antibacterial activity	[62,63,64]
*P. oxalicum*	Polygalacturonase, pectin lyase, pectinesterase; Cellulase, xylanase and feruloyl esterase; Polyphenolic compounds; Organic acids	Clarification and Depectinization of juice; Ferulic acid production; Antioxidant and genoprotective activity. Biofertilizer	[65,66,67,68]
*P. dipodomyicola*	Peniphenone B–C; β-glucosidase	Antibacterial activity; Biotransformation of ginsenoside	[69,70]

## Data Availability

Please contact author for data request.

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
