# Peer review of "Isolation and Molecular Characterization of Indigenous Penicillium chrysogenum/rubens Strain Portfolio for Penicillin V Production"

_microorganisms, 2023, doi:10.3390/microorganisms11051132_

Round 1

Reviewer 1 Report

It is difficult to read manuscript due to poor English.- It would be useful to correct the language to improve  the quality of English in the publication

Introduction:

- please reframe sentence „that can segregate tirelessly allied species”

- please correct ‘strep throt’ to ‘strep throat’

- ‘Wide spread uses of these antibiotics, many microbes developed antimicrobial resistance (AMR), is an emerging issue globally, penicillin plays an important role.’ - sentence needs to be corrected.
In general, many sentences in the manuscript should be improved in terms of linguistic correctness, I will not point out on the remaining sentences, but linguistic correction should be done.

- ITS is not a gene name and ‘BenA’ as a gene should be italicized

- Section 2.5 - sample preparation has been described in detail, while the actual performance of profiling with LC-HRMS is not described at all, LC-HRMS methodology should be added.

- Section 2.7 - No description of the methodology for HPLC (if a methodology from a publication is used, a citation and at least a brief description would be useful). Or did you use the method for simultaneous detection of PenV, POA, and 6-APA from section 2.10?

- Section 2.9 - The control of Pen activity against the indicator strain is obviously necessary, but I don't quite understand why the concentration of PenV obtained was determined based on the zone of inhibition compared to the Pen standard instead of quantitative HPLC analysis? This is not a very accurate method, even if replicates were performed (of which nothing was mentioned) and more than one bacterial strain was used (which would be desirable).

- Section 2.10 Have matrix effects been determined to assess the reliability and selectivity of the method?

Figure 2 - illegible, I suggest expanding to a full page

-Section 3.4. - Why did the authors decided on Metabolic profiling by LC-HRMS in only one strain among P. chrysogenum and P. rubens? It would have been more beneficial to study a larger group, at least three representatives of each strain. If these molecules are intended to be used for distinguishing closely related species then it would be beneficial to compare a larger group and show that there is reproducibility in this regard within a strain.

-I would suggest expanding the discussion section, as it is poorly executed.

Reviewer 2 Report

Sawant et al.  cultivated Penicillium spp., confirmed by and produced Beta (β)-lactam antibiotics. The study was organized well, with a promising exciting hypothesis. The study was designed well and presented well. Therefore, it is recommended for accepted for publishing.  I hope Figures 2 & 3 will be organized in good resolution in the published version.

Round 2

Reviewer 1 Report

Please correct further typos, minor linguistic errors and correct that the benA gene is italicized everywhere. Also, I would suggest you rethink improving the readability of Figure 2 and Figure 3.